# Evaluating antibody mediated opsonophagocytosis of bacteria via lab protocol: RAW 264.7 cell phagocytosis assay

**Matthew Slarve**[1], **Travis Nielsen**[1], **Dongran Yu**[1], **Jordyn VanOrman**[1], **Brian Luna**[1], **Brad Spellberg**[2]

**1** Molecular Microbiology and Immunology Department, University of Southern California, Los Angeles California, United States of America, **2** Los Angeles General Medical Center, Los Angeles, California, United States of America

## Abstract

Assessing the phagocytosis of microbes by macrophages is an important component of studies of novel immunotherapeutics, antimicrobial drugs, immune effectors, or any immunology related research. Here we define two protocols for measuring *in vitro* phagocytosis by RAW 246.7 cells – a photographic phagocytosis assay that allows optical measurement of bacterial cells inside of the RAW 246.7 cell by staining fixed cells and visually quantifying the bacteria that have been phagocytosed, as well as a killing assay, which measures the viable bacteria released from lysed cells following exposure and phagocytosis via colony forming unit analysis. These methods differ from previously available protocols for measuring phagocytosis, as they are more generalizable to a variety of microbes and experimental designs and use readily available cell lines and materials to produce robust results. We demonstrate the utility of our methods by showing how opsonization by our novel, therapeutic monoclonal antibodies drive phagocytosis and killing of the gram-negative bacteria, *Acinetobacter baumannii*.

## Introduction

Phagocytosis is a critical step in immunity against microbial pathogens, serving as an early innate response to infection. Phagocytes such as macrophages may interact directly with pathogen-associated molecular patterns, or with opsonins such as antibodies or complement that have bound to the microbial surface, triggering the envelopment of the pathogen in a phagosome [1]. This phagosome then matures into a phagolysosome following a series of fusions and fissions of endocytic vesicles, and the final joining of a lysosome [2]. The phagolysosome acts as a microbicidal organelle, supplying the necessary hydrolytic enzymes, pH, and reactive oxygen species needed to secure the destruction of the microbial invader [3]. This vital immunological process is useful in research as a measure of innate immunity. Reliable and

**Data availability statement:** All relevant data are within the paper and its Supporting information files.

**Funding:** This work was funded by NIAID grant 2R01 AI130060 to B.S.

**Competing interests:** The authors have declared that no competing interests exist.

consistent measurements of phagocytosis require a standardized protocol that yields reproducible results with minimal training and cost. Currently available protocols often describe *ex vivo* methods that measure the phagocytosis of primary cells, or require fluorescently labeled bacterial inocula [4,5]. These protocols have distinct uses, but have limited applicability to broad, high throughput *in vitro* studies. Furthermore, while use of primary cells has advantages, collection of primary cells for *ex vivo* assays is costly, can require long turn-around times between individual experiments, and introduces considerable variability in macrophage functions depending on the donor. In contrast, use of a single macrophage cell line eliminates donor to donor variability, is less expensive, more facile, and enables higher throughput investigations. Phagocytosis assays using fluorescently labeled bacteria are unlikely to be representative of clinical microbial isolates and require expensive equipment to analyze.

Here we present an inexpensive, generalizable protocol that is useful for most any clinically important microbes, requires minimal equipment, and is an easily measurable medium-throughput readout for evaluating phagocytosis. The first method described in the protocol is photographic, in which cells are fixed, stained, and photographed to enable visual counting of bacteria and interpretation of phagocytosis as the number of bacteria that have been visibly taken up by individual macrophages in the photo. The second method describes colony forming unit analysis of lysed macrophages to determine the amount of viable bacteria that were present in the macrophage at a specific timepoint. These methods have been extensively validated via our lab's publications studying immunotherapeutic solutions to drug-resistant bacterial infections [6–12], and have thus become a mainstay in our lab's investigation of therapeutic monoclonal antibodies (MAbs) against *A. baumannii*. Two such MAbs are disclosed here for the first time and were both produced via target-agnostic methods. The *in vitro* phagocytosis assay protocol described here demonstrates the promising opsonization capabilities of these MAbs, advocating for their further preclinical development.

## Methods

The protocol described in this peer-reviewed article is published on protocols.io, (DOI: dx.doi.org/10.17504/protocols.io.yxmvmmr16v3p/v1) and is included for printing as supporting information file 1 with this article.

### Experimental controls:

The specific experiments shown here included only negative controls given that no clear positive control for this design exists. In lieu of our experimental monoclonal antibodies, negative control groups received an equal dose of IgG isotype control antibody (Fisher Scientific, cat# MAB002), which was designed for use as a control in flow cytometry or ELISA assays. For other experimental designs testing immunostimulatory compounds rather than MAbs, our lab has used naïve cells that were not activated with IFN-γ as a negative control given their low propensity for phagocytosis, and IFN-γ treated cells as a positive control [11].

## Reproducibility:

We would recommend 2–3 biological replicates per condition (IE- two to three wells devoted to the same experimental groups), and for each assay to be done in duplicate. We have found that these conditions promote strong inter-assay reproducibility.

## Expected results

This protocol can be used to visualize the uptake of bacteria by RAW 264.7 cells and can demonstrate the direct effect of various experimental drugs or compounds on phagocyte activity (Fig 1). Our lab uses this method as an *in vitro* screening step for our therapeutic MAb development, as we have found that our MAbs direct phagocytosis *in vivo*. Some important considerations for carrying out these experiments are that the timing of the staining step is critical and can have major implications for data interpretation. If under-stained, it may be difficult to see the features of an individual macrophage (Fig S1A). If over-stained, the cells may appear too dark to easily view the bacteria within (Fig S1B).

Additionally, the data can be heavily biased by the imaging process. Therefore, to avoid potential bias, it is advisable to blind the image acquisition process by having the photos taken by a second researcher who is blinded to the experimental

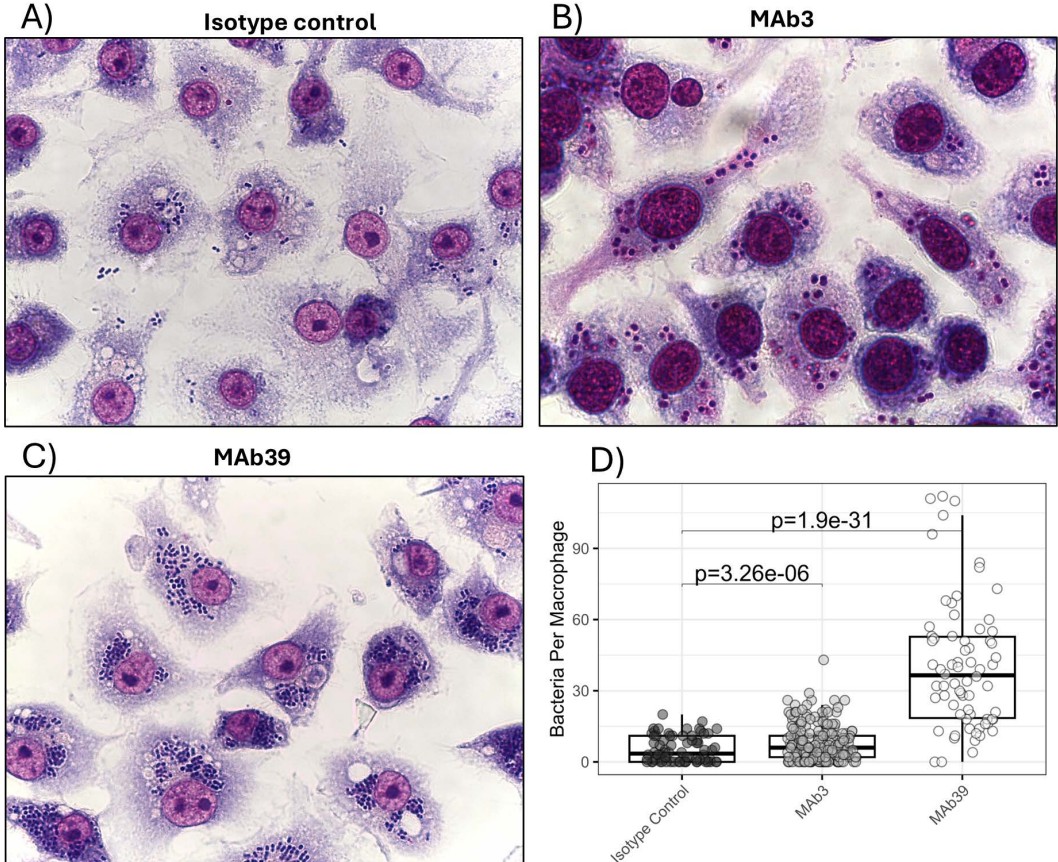

**Fig 1. Photographic phagocytosis assay.** RAW 246.7 cells were exposed to *A. baumannii* with a multiplicity of infection of 1:20. Cells were given 10 µg/mL of either a non-specific IgG isotype control antibody (A), or one of two experimental MAbs, MAb3 (B) and MAb39 (C) in the presence of 10% CD1 mouse serum. Cells were stained and photographed, and phagocytosis was evaluated using Fiji software. Non-parametric statistical analyses were performed via Mann-Whitney U test (D).

groups. Furthermore, a blinded researcher should examine the entire slide and ensure that the fields they are photographing are representative of the slide as a whole, not just photographing the areas of highest or lowest phagocytosis. Additionally, it is important to only include the macrophages that are fully visible within the photo and exclude any that are clipped by the edge of the visible field during the cell counting process.

The data can then be expressed as the number of bacteria that have been phagocytosed by each RAW 264.7 cell, with a single datum representing an individual macrophage (Fig 1D). Due to normal variance of phagocytosis, we have found it necessary to collect data from 50–100 individual macrophages. However, these relatively high n-values allow for observing even modest changes as highly statistically different. Therefore, it remains critical to assess if modest changes in phagocytosis result in meaningful changes in a whole animal infection model. Fig 1D shows an example of this phenomenon where our experimental antibody MAb3 provided only modest improvement in phagocytosis compared to the isotype control despite a strong statistical significance, while MAb39 provided a much more impressive effect.

Interpretation of the kill assay data is more straightforward and easily examined as CFU/mL of bacteria in the volume of cell lysate (Fig 2). Given that the kill assay measures any living bacteria that were present in the well following washing, it can be useful to kill any non-phagocytosed bacteria using a gentamicin treatment (optional step #6). Gentamicin does not penetrate the macrophage quickly, a brief incubation with this antibiotic will kill bacteria that are outside the macrophages but not harm those that have been phagocytosed, resulting in cleaner, more accurate bacterial counts. Additionally, data from this assay is best taken in the context of the photographic phagocytosis assay, as phagocytosis and intracellular killing are independent processes and an increase in phagocytosis may not always correlate with increased intracellular killing.

Some potential limitations to the described protocol are that it is optimized for a specific cell line – RAW 264.7 cells, which are mouse macrophages. Investigators who are interested in using different phagocytes (dendritic cells, neutrophils, etc.), primary cells, or cell lines derived from a different animal may need to modify the procedure to fit their experimental needs. Furthermore, though this protocol can be used with numerous different microbial pathogens, we have found that some microbes (particularly avirulent bacteria) are so easily taken up by macrophages that no treatment shows significant increase in phagoctytosis. Therefore, new isolates must be screened in a pilot study before committing to a more meaningful experiment.

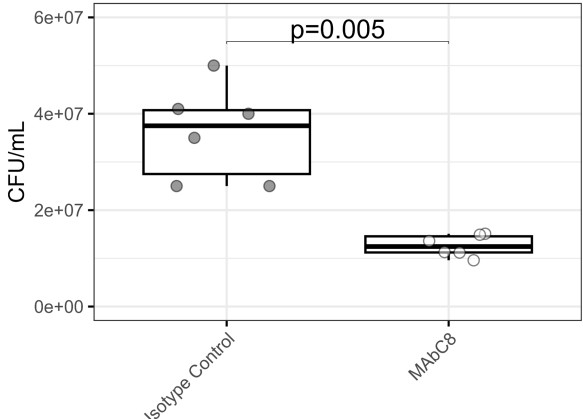

**Fig 2. Phagocytosis-Kill assay.** RAW 246.7 cells were exposed to *A. baumannii* with a multiplicity of infection of 1:20. Cells were given 10 μg/mL of either a non-specific IgG isotype control antibody or our experimental antibody, MAbC8. Following phagocytosis, RAW 246.7 cells were lysed, and *A. baumannii* within was collected and quantitatively plated to determine CFU/mL lysate. Non-parametric statistical analyses were performed via Mann-Whitney U test. *This data previously published* [7].

Our lab has found that with proper implementation, this assay supplies an important indicator in testing new antimicrobial monoclonal antibodies and immunostimulatory compounds.

## Supporting information

**S1 Protocol. RAW 264.7 cell phagocytosis assay protocol.** The protocol described in this peer-reviewed article is published on protocols.io, (DOI: dx.doi.org/10.17504/protocols.io.yxmvmmr16v3p/v1).
(PDF)

**S1 Fig. Examples of improperly stained phagocytosis assays.** RAW 246.7 cells were exposed to A. baumannii with a multiplicity of infection of 1:20. Phagocytosis, washing, and fixing steps were performed as according to the protocol, but staining was not. (A) Cells were stained for only 10 seconds per stain, as opposed to the appropriate 1 minute for HEMA stain I and 45 seconds for HEMA stain II as listed in the protocol. (B) Cells were stained for one minute and 30 seconds per stain as opposed to the appropriate times listed in the protocol and above. These assays are unreadable.
(TIF)

## Author contributions

**Conceptualization:** Matthew Slarve, Brian Luna.

**Data curation:** Matthew Slarve, Travis Nielsen, Dongran Yu.

**Formal analysis:** Matthew Slarve.

**Funding acquisition:** Brad Spellberg.

**Investigation:** Matthew Slarve, Travis Nielsen, Jordyn VanOrman, Dongran Yu.

**Methodology:** Matthew Slarve, Travis Nielsen, Jordyn VanOrman.

**Software:** Matthew Slarve.

**Supervision:** Brian Luna.

**Visualization:** Matthew Slarve.

**Writing – original draft:** Matthew Slarve.

**Writing – review & editing:** Matthew Slarve, Brian Luna, Brad Spellberg.

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
