## [Decision Letter · Decision Letter 0]

9 Jul 2025

PONE-D-25-29912Evaluating antibody mediated opsonophagocytosis of bacteria via RAW 264.7 cell uptake assaysPLOS ONE

Dear Dr. Luna,

Thank you for submitting your manuscript to PLOS ONE. After careful consideration, we feel that it has merit but does not fully meet PLOS ONE’s publication criteria as it currently stands. Therefore, we invite you to submit a revised version of the manuscript that addresses the points raised during the review process.

**Based on reports I received from our evaluators, I have decided that your manuscript requires a major revision. Please consider all detailed comments, as provided by reviewers, and resubmit the revised version. **

We look forward to receiving your revised manuscript.

Kind regards,

Mohammad Faezi Ghasemi, Ph.D

Academic Editor

PLOS ONE

Journal Requirements: 

 [NIAID 2R01 AI130060 to B.S.]. 

4. We note you have not yet provided a protocols.io PDF version of your protocol and/or a protocols.io DOI. When you submit your revision, please provide a PDF version of your protocol as generated by protocols.io (the file will have the protocols.io logo in the upper right corner of the first page) as a Supporting Information file. The filename should be S1_file.pdf, and you should enter “S1 File” into the Description field. Any additional protocols should be numbered S2, S3, and so on. Please also follow the instructions for Supporting Information captions [https://journals.plos.org/plosone/s/supporting-information#loc-captions]. The title in the caption should read: “Step-by-step protocol, also available on protocols.io.”

Please assign your protocol a protocols.io DOI, if you have not already done so, and include the following line in the Materials and Methods section of your manuscript: “The protocol described in this peer-reviewed article is published on protocols.io (https://dx.doi.org/10.17504/protocols.io.[...]) and is included for printing purposes as S1 File.” You should also supply the DOI in the Protocols.io DOI field of the submission form when you submit your revision.

If you have not yet uploaded your protocol to protocols.io, you are invited to use the platform’s protocol entry service [https://www.protocols.io/we-enter-protocols] for doing so, at no charge. Through this service, the team at protocols.io will enter your protocol for you and format it in a way that takes advantage of the platform’s features. When submitting your protocol to the protocol entry service please include the customer code PLOS2022 in the Note field and indicate that your protocol is associated with a PLOS ONE Lab Protocol Submission. You should also include the title and manuscript number of your PLOS ONE submission.

Reviewers' comments:

Reviewer's Responses to Questions

**Comments to the Author**

1. Does the manuscript report a protocol which is of utility to the research community and adds value to the published literature?

Reviewer #1: Yes

Reviewer #2: Yes

2. Has the protocol been described in sufficient detail?

To answer this question, please click the link to protocols.io in the Materials and Methods section of the manuscript (if a link has been provided) or consult the step-by-step protocol in the Supporting Information files.

The step-by-step protocol should contain sufficient detail for another researcher to be able to reproduce all experiments and analyses.

Reviewer #1: Partly

Reviewer #2: Partly

3. Does the protocol describe a validated method?

Reviewer #1: No

Reviewer #2: Yes

4. If the manuscript contains new data, have the authors made this data fully available?

Reviewer #1: Yes

Reviewer #2: Yes

**5. Is the article presented in an intelligible fashion and written in standard English?**

Reviewer #1: Yes

Reviewer #2: Yes

6. Review Comments to the Author

Reviewer #1: Dear Author,

This manuscript is potentially acceptable pending the resolution of major issues outlined below. Your study on measuring phagocytosis by RAW 246.7 cells and evaluating the effect of monoclonal antibodies on Acinetobacter baumannii addresses an important and practical topic in immunology and pharmacology. We appreciate your effort in this valuable research. However, we suggest the following points to improve your manuscript:

1. Please provide more detailed explanations of the phagocytosis mechanism and its biological stages to help readers better understand the underlying processes.

2. Include information about experimental controls and the reproducibility of your results in the methods section to enhance the scientific rigor of your work.

3. Discuss the limitations and potential challenges of the methods you introduced to make the manuscript more comprehensive and transparent.

4. Consider expanding the discussion to include the applicability of these methods to other microorganisms or at least mention this aspect to broaden the scope of your research.

5. Review and refine the sentence structure and writing style of the abstract to make it clearer, more fluent, and more suitable for a specialized audience.

6. Figure 1 does not have sufficient transparency and clarity; please improve its quality.

7. The full title is identical to the short title and should be corrected to better reflect the content.

8. The description of experimental procedures lacks sufficient detail. Please provide comprehensive information on cell culture conditions, bacterial concentrations, incubation times, and sample preparation steps.

9. It is important to include descriptions of both negative and positive controls to validate your results and ensure the reliability of the assays.

10. The criteria and quantitative measures used for assessing phagocytosis and bacterial killing in both the imaging and killing assays need to be clearly defined.

11. Please specify environmental conditions such as temperature and pH, as well as details about the equipment used (e.g., microscope type, colony counting devices) to facilitate accurate replication.

12. Discuss any limitations or potential challenges associated with the methods to provide a balanced perspective.

13. Indicate the number of independent experimental replicates performed to support the statistical validity of your findings.

14. Addressing these points will significantly strengthen the methods section and improve the overall quality of your manuscript.

We hope that by addressing these points, your manuscript will be well-prepared for publication in PLOS ONE. We look forward to reviewing your revised manuscript and wish you success.

Best regards

Reviewer #2: -I recommend revising the title to specify the lab protocol in title.

-Add methodology and results to Abstract. Compare the results obtained from two used method.

-Introduction is so brief. Explain more about currently available protocols, introduce the novel monoclonal antibodies (their characteristics, advantages, …).

-List the MABs in materials (table) in protocol.

-In protocol, mention the CFU/mL of 50 uL bacterial suspension.

-It is recommended to add figure for results represented in lines 46-50.

7. PLOS authors have the option to publish the peer review history of their article (what does this mean? ). If published, this will include your full peer review and any attached files.

**Do you want your identity to be public for this peer review?** For information about this choice, including consent withdrawal, please see our Privacy Policy .

Reviewer #1: **Yes: ** no

Reviewer #2: No

---

## [Author Response · Author response to Decision Letter 1]

14 Aug 2025

Dear Reviewers,

Thank you for your constructive feedback on our manuscript and protocol for this lab protocol submission to PLOSone. We feel that your comments have been excellent in improving the quality of this submission, and we hope that the changes we have made to address your concerns have been satisfactory. Below, please find a point-by-point breakdown of our response to each of the comments left by both reviewers. Thank you for your contribution.

1. Please provide more detailed explanations of the phagocytosis mechanism and its biological stages to help readers better understand the underlying processes.

- A brief introduction to the fundamentals of phagocytosis has been added, including citations of papers with detailed descriptions of the process for interested readers.

2. Include information about experimental controls and the reproducibility of your results in the methods section to enhance the scientific rigor of your work.

- These sections have been added to the methods section of the main manuscript

3. Discuss the limitations and potential challenges of the methods you introduced to make the manuscript more comprehensive and transparent.

- Limitations and potential challenges have been added to the conclusion

4. Consider expanding the discussion to include the applicability of these methods to other microorganisms or at least mention this aspect to broaden the scope of your research.

- A statement has been added to the conclusion that indicates that this assay works for other microbes.

5. Review and refine the sentence structure and writing style of the abstract to make it clearer, more fluent, and more suitable for a specialized audience.

- The abstract has been thusly adjusted

6. Figure 1 does not have sufficient transparency and clarity; please improve its quality.

- The older figure has been replaced with a higher resolution image.

7. The full title is identical to the short title and should be corrected to better reflect the content.

- The titles have been updated and are now different.

8. The description of experimental procedures lacks sufficient detail. Please provide comprehensive information on cell culture conditions, bacterial concentrations, incubation times, and sample preparation steps.

- The protocol has been updated to clearly specify the culture and incubation conditions, concentration of the bacterial inoculum, and all preparation of samples.

9. It is important to include descriptions of both negative and positive controls to validate your results and ensure the reliability of the assays.

- A description of negative and positive controls has been added to the methods section of the manuscript, and as an addendum to the end of the protocol.

10. The criteria and quantitative measures used for assessing phagocytosis and bacterial killing in both the imaging and killing assays need to be clearly defined.

-These descriptions are now included in both the methods and expected results sections of the main manuscript.

11. Please specify environmental conditions such as temperature and pH, as well as details about the equipment used (e.g., microscope type, colony counting devices) to facilitate accurate replication.

- The protocol has been updated to specify the culture conditions, as well as the microscope and camera that our lab uses.

12. Discuss any limitations or potential challenges associated with the methods to provide a balanced perspective.

- This has been added to the conclusion section of the manuscript

13. Indicate the number of independent experimental replicates performed to support the statistical validity of your findings.

- A section has been added to the methods section that addresses this concern

Reviewer #2: -I recommend revising the title to specify the lab protocol in title.

-The title has been thusly modified.

-Add methodology and results to Abstract. Compare the results obtained from two used method.

- The abstract has been enhanced with a brief description of the methodology, and explanation of how our methods differ from previous protocols.

-Introduction is so brief. Explain more about currently available protocols, introduce the novel monoclonal antibodies (their characteristics, advantages, …).

- The introduction has been enhanced with additional information about the other available protocols, and some context about the novel MAbs.

-List the MABs in materials (table) in protocol.

- Because this protocol is meant to cover more than just monoclonal antibody investigations, we instead updated the table to include any experimental substance being used by the investigator.

-In protocol, mention the CFU/mL of 50 uL bacterial suspension.

- the protocol has been updated to fix this oversight.

-It is recommended to add figure for results represented in lines 46-50.

- Examples of improperly stained figures have been added as supplemental figure S1. These new supplementary figures have also been referenced in the text at the aforementioned lines.

---

## [Decision Letter · Decision Letter 1]

17 Aug 2025

Evaluating antibody mediated opsonophagocytosis of bacteria via lab protocol: RAW 264.7 cell phagocytosis assay

PONE-D-25-29912R1

Dear Dr. Luna,

We’re pleased to inform you that your manuscript has been judged scientifically suitable for publication and will be formally accepted for publication once it meets all outstanding technical requirements.

Kind regards,

Mohammad Faezi Ghasemi, Ph.D

Academic Editor

PLOS ONE

Additional Editor Comments (optional):

Reviewers' comments:

Reviewer's Responses to Questions

**Comments to the Author**

1. Does the manuscript report a protocol which is of utility to the research community and adds value to the published literature?

Reviewer #1: Yes

2. Has the protocol been described in sufficient detail?

To answer this question, please click the link to protocols.io in the Materials and Methods section of the manuscript (if a link has been provided) or consult the step-by-step protocol in the Supporting Information files.

The step-by-step protocol should contain sufficient detail for another researcher to be able to reproduce all experiments and analyses.

Reviewer #1: Yes

3. Does the protocol describe a validated method?

Reviewer #1: Yes

4. If the manuscript contains new data, have the authors made this data fully available?

Reviewer #1: Yes

**5. Is the article presented in an intelligible fashion and written in standard English?**

Reviewer #1: Yes

6. Review Comments to the Author

Reviewer #1: Given that the authors have resolved the issues as much as possible, and the subject of this article is innovative, this article can be accepted and pave the way for other researchers' research.

7. PLOS authors have the option to publish the peer review history of their article (what does this mean? ). If published, this will include your full peer review and any attached files.

**Do you want your identity to be public for this peer review?** For information about this choice, including consent withdrawal, please see our Privacy Policy .

Reviewer #1: **Yes: ** Dr.Mohaddeseh Larypoor

---

## [Editor Report · Acceptance letter]

PONE-D-25-29912R1

PLOS ONE

Dear Dr. Luna,

I'm pleased to inform you that your manuscript has been deemed suitable for publication in PLOS ONE. Congratulations! Your manuscript is now being handed over to our production team.

Kind regards,

on behalf of

Dr. Mohammad Faezi Ghasemi

Academic Editor

PLOS ONE